# Metrics for Time-to-Event Prediction of Gaze Events

**Tim Rolff**                    Tim.Rolff@uni-hamburg.de  *Universität Hamburg*
**Niklas Stein**                      niklas.stein@wwu.de  *University of Münster*
**Markus Lappe**                  mlappe@uni-muenster.de  *University of Münster*
**Frank Steinicke**          Frank.Steinicke@uni-hamburg.de  *Universität Hamburg*
**Simone Frintrop**          Simone.Frintrop@uni-hamburg.de  *Universität Hamburg*

## Abstract

In this paper, we explore metrics for the evaluation of time-to-saccade problems. We define a new sampling strategy that takes the temporal nature of gaze data and time-to-saccade problems into account, avoiding samples of the same event in different datasets. This allows us to define novel error metrics for a more intuitive evaluation of predicted durations. The metrics are defined to evaluate the consistency of a predictor and the evaluation of the error over time. We evaluate our method using a state-of-the-art method for time-to-saccade prediction along with an average baseline on three different datasets.

**Keywords:** time-to-saccade prediction, gaze classification, time-to-event, metrics

## 1. Introduction

When we as humans perceive a scene, our eyes constantly move due to their relatively small area of sharp vision, the fovea Holmqvist et al. (2011). This restriction also applies when we perceive a virtual environment (*VE*) through a head-mounted display (*HMD*). In both cases, it is possible to determine the different eye-movements by taking their inherent properties, such as velocity and acceleration, into account and classify them into their respective class Komogortsev and Karpov (2013); Andersson et al. (2017); Startsev et al. (2019); Zemblys et al. (2019); Agtzidis et al.; Salvucci and Goldberg (2000); Dar et al. (2021). However, such a classification can only be performed after capturing the sample. This, makes a real-time utilization of gaze events or blinks challenging due to the low update rates and high latencies found in commercial head-mounted displays or wearable eye-trackers Stein et al. (2021); Langbehn et al. (2018). This is especially true for saccades, as they are temporarily short, fast eye-movements in the range of 30-80 ms Holmqvist et al. (2011), where wearable eye-trackers often just have a few samples to classify them correctly.

Nonetheless, knowing when a saccade event occurs would benefit several virtual reality (*VR*) applications, such as gaze forecasting Hu et al. (2020, 2021), blink or saccade detection for redirected walking Langbehn et al. (2018); Sun et al. (2018), gaze contingent rendering Arabadzhiyska et al. (2017), or gaze-based interaction David-John et al. (2021). Furthermore, the prediction of fixation durations is also important in other areas outside *VR*, with one example being scan path prediction which try to predict fixation durations along with the sequence of fixation points on a visual stimulus Yang et al. (2020). To use these gaze events, previous applications often mitigate the latency through unnatural actions, such as intentional blinking Langbehn et al. (2018) or long saccade durations Sun et al. (2018).

A different approach was recently proposed by Rolff et al. (2022). They redefined the problem of gaze classification as the recurrent time-to-event prediction of saccade events (*time-to-saccade*), predicting the time it takes until a saccade occurs. However, this approach is fairly general and can also be applied to other gaze events, such as fixations or blinks. In contrast to classical gaze classification approaches, this redefinition of gaze event classification as a recurrent time-to-event problem allows estimating the remaining time for each input sample of an eye-tracker. This provides information on how long it will take until the specified event will occur. This is desirable, as it is not essential if the class for each time-step is known, but rather when its class will change. In contrast to classical gaze classification methods, the redefinition also allows to account for the latency of eye-trackers found in commercial head-mounted displays or wearable eye-trackers. To evaluate their approach, Rolff et al. (2022) utilize the mean absolute error (*mae*) on a set of randomly sampled time-to-event values to evaluate how well their method performs for time-to-saccade prediction.

In this paper, we define a more fitting sampling strategy than random sampling. This allows to adapt the previously used error metric to be more suited to the actual problem of time-to-saccade prediction. We will explore how well these metrics can be utilized to understand the prediction and provide a different evaluation method.

To summarize, our work proposes the following contributions:

- A sampling strategy for dataset sampling that considers the sequential time-to-event information of gaze events.

- Novel error metrics using the previously defined sampling strategy, enabling a more interpretable result to infer the predictive performance of a time-to-saccade predictor.

## 2. Related Work

A commonly utilized metric for time-to-event problems is Harrell's concordance index (*c-index*) Harrell et al. (1982). The *c-index* measures the correlation between the predicted risk-score and the observed time-to-event. Hence, a higher risk value correlates with a shorter time-to-event. However, it has also been shown that the *c-index* is biased if the test set contains a high number of censored samples Uno et al. (2011), leading to an alternative definition by Uno et al. (2011). Another metrics commonly used is the brier-score Brier (1950). Its definition is equivalent to the mean square error (*mse*) for probabilistic predictions of binary events, hence, requiring a probabilistic prediction from the employed model. It has also been re-defined to allow censored data Graf et al. (1999).

Besides survival-analysis related metrics, there are multiple metrics for time-series forecasting using the real values of the prediction. Commonly used metrics are the mean-absolute, mean-square, or (normalized) root-mean-square error Diebold (1998); Hyndman and Koehler (2006). Variations of those metrics have been proposed, such as the (symmetric) mean absolute percentage error. Most of the listed error metrics, assume that an overestimation should be penalized equally. This, however, might not always be the case, thus requiring an asymmetric error function such as the asymmetric mean Diebold (1998) or the linex error Diebold (1998); Varian (1975).

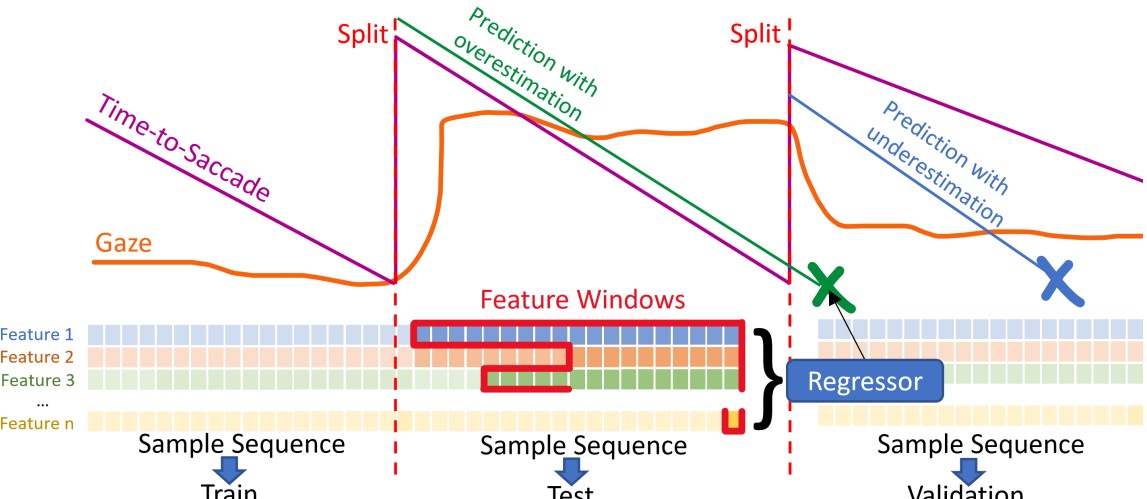

Figure 1: Illustration of the sampling strategy, splitting the data at the occurrence of an event into multiple sequences (red). Each sequence then contains multiple samples and is placed into the respective dataset. It also depicts an example for a prediction with overestimation (green) and a prediction with underestimation (blue).

## 3. Methodology & Metrics

For our experimental setup, we follow, if not noted otherwise, Rolff et al. (2022), allowing for comparability between both approaches. First, we would like to highlight a disadvantage of their evaluation, the random sampling. This does not take the temporal property of the gaze data into account, as the samples of the same time-to-saccade sequence might have been selected for different datasets. As a result, it is impossible to evaluate properties, like the consistency of the prediction, or the error of the overall sequence, without the predictor having seen part of the data. This makes it challenging to interpret the reported error metrics, as it is not clear how the predictor behaves over time.

Here, we introduce a new sampling strategy that keeps samples of the same time-to-saccade sequence in the same dataset. As illustrated in Fig. 1, we construct these sequences, by splitting the gaze signal exactly if an event happens, instead of randomly chosen samples. This results in sequences containing multiple individual samples as data points. An advantage is that the time-to-event of a sequence is always strictly monotonically decreasing, with the rate depended on the frequency of the eye-tracker. With the update frequency $f_i > 0$ of the eye-tracker at step $i$, each time-to-saccade value (tts) can be calculated through: $\text{tts}_{i+1} = \text{tts}_i - f_i^{-1}$. We define the last time-to-event before the occurrence of the event as zero, implying the first time-to-event value being equal to the total duration of the sequence.

Using these observations, it allows us to explore additional error metrics that account for the temporal properties of a time-to-saccade sequence which also take and information of eye-movement classifiers into account. As time-to-saccade predictions are rarely right censored, as they are repeatable events that happen every 300 to 2500 ms. Therefore, same as Rolff et al. (2022) we would like to advise against the usage of earlier listed time-to-event

metrics for the evaluation of predictions, even under the new sampling strategy. As a result, the only right censored sequences are at the end of an eye-tracking session, often corresponding to a small portion of the dataset. Moreover, the task of time-to-saccade prediction requires predicting the time-to-event as accurate as possible, other metrics such as the *c-index* do not provide helpful information on their accuracy. Here, it is better to utilize metrics for time-series forecasting, as they are concerned with the difference between the actual time-to-event and the predicted time. However, as these are fairly general and do not allow insight into a time-to-saccade predictor model, we propose the additional metrics:

**Consistency:** To measure how consistent the model is in its prediction, we define consistency of a sequence $j$ with length $l$ as the relative difference Diebold (1998) between the current and the next prediction. Ideally, this change should be equal to the frequency of the eye-tracker, due to the definition of time-to-saccade. Hence, we can define *consistency* as:

$$c_j = \sum\nolimits_{i=0}^{l-1} \frac{\left| |p_{i+1} - p_i| - f_{i+1}^{-1} \right|}{|f_{i+1}^{-1}|}.$$

As this gets evaluated over each sequence, we can derive the mean consistency of a dataset through the arithmetic mean.

**Average overestimation and underestimation rate:** Overestimation and underestimation measure if the model generally tends to predict durations that are too short (*underestimation*) or too long (*overestimation*). For the non-temporal time-to-saccade problem of a sample $j$ with time-to-saccade duration $d_j$ and predicted duration $p_j$, an overestimation happens if $d_j - p_j < 0$, and underestimation if $d_j - p_j > 0$. This calculation is not possible with a recurrent time-to-saccade prediction, as the predictor outputs an estimation $p_{j_i}$ for each step $i$. Therefore, we calculate the average time-to-saccade $p_j$ using the arithmetic mean for the estimation of over- and underestimation of the sequence $j$. While this is not optimal as the prediction might over- or underestimations with time, we assume this to be a reasonable approximation for a general overview. This allows us to define the average overestimation and underestimation rate for a set of $n$ sequences as:

$$\text{avg. overestimation rate} = \frac{1}{n}\sum\nolimits_{j=1}^{n} \mathbb{1}_{d_j < p_j}, \quad \text{avg. underestimation rate} = \frac{1}{n}\sum\nolimits_{j=1}^{n} \mathbb{1}_{p_j < d_j}$$

**Average sequence and underestimation error:** Using the historic gaze information provided by the eye-tracker, we would not perform an action in case of an overestimation. This is not the case for an underestimation, as we cannot exploit the additional information that would imply a wrongfully performed action. Hence, we calculate the underestimation error only in cases where the prediction of a model underestimates the actual duration, and assume a perfect prediction otherwise, by defining the average underestimation error as:

$$\text{avg. underestimation error}_f = \frac{1}{n}\sum\nolimits_{j=1}^{n} f(d_j, p_j) \cdot \mathbb{1}_{p_j < d_j},$$

using the indicator function $\mathbb{1}$ and the error metrics $f \in \{\text{mse}, \text{mae}\}$. To calculate the average time-to-saccade error (avg. tts.), we use previous definitions of time-to-saccade duration $d_j$ and average time-to-saccade prediction $p_j$ of a sequence $j$:

$$\text{avg. sequence error}_f = \frac{1}{n}\sum\nolimits_{j=1}^{n} f(d_j, p_j)$$

Table 1: Results of the stochastic gradient descent (SGD) regressor and average time-to-event using the metrics described in Sec. 3 along with the mean square error (mse) and mean absolute error (mae). A lower error is preferred.

| Metric / Dataset | Stochastic Gradient Descent | | | Avg. Time-to-Event | | |
|---|---|---|---|---|---|---|
| | DGaze | FixationNet | EGTEA | DGaze | FixationNet | EGTEA |
| mse↓ | 0.1285 $s^2$ | 0.2390 $s^2$ | 0.1672 $s^2$ | 0.2314 $s^2$ | 0.3647 $s^2$ | 0.3043 $s^2$ |
| mae↓ | 0.2556 s | 0.3567 s | 0.2668 s | 0.3387 s | 0.4261 s | 0.3677 s |
| avg. time-to-saccade mse↓ | 0.0494 $s^2$ | 0.0887 $s^2$ | 0.0420 $s^2$ | 0.0672 $s^2$ | 0.1035 $s^2$ | 0.0745 $s^2$ |
| avg. time-to-saccade mae↓ | 0.1747 s | 0.2422 s | 0.1484 s | 0.1792 s | 0.2260 s | 0.1765 s |
| underestimation mse↓ | 0.0319 $s^2$ | 0.0537 $s^2$ | 0.0311 $s^2$ | 0.0664 $s^2$ | 0.1003 $s^2$ | 0.0718 $s^2$ |
| underestimation mae↓ | 0.0857 s | 0.1105 s | 0.0799 s | 0.1666 s | 0.1968 s | 0.1470 s |
| o/u estimation rate↓ | 0.61/0.39 | 0.64/0.36 | 0.60/0.40 | 0.27/0.73 | 0.36/0.64 | 0.43/0.57 |
| consistency↓ | 1.64 | 1.39 | 1.12 | 1.0 | 1.0 | 1.0 |

**Sectioning:** The prediction of a model may change with time. Depending on the utilized method, it might not have enough information at the beginning to predict an accurate time-to-saccade. As a result, the prediction may improve over time without being inherently evident from the evaluation when using the earlier mentioned metrics. Hence, we split each time-to-saccade sequence $S_j[1, \ldots, l_j]$ of length $l_j$ into $k$ sections $s_{j_k} = S_j[\lceil \frac{l_j}{k} \cdot (k-1) \rceil, \ldots, \lfloor \frac{l_j}{k} \cdot k \rfloor]$. Then we calculate the error over all sections $S_m = \{s_{0_m}, \ldots, s_{n_m}\}$ of the same bin $m$, showing the behavior of the error over time.

## 4. Evaluation & Discussion

To evaluate our approach defined in Sec. 3 we utilize a linear regressor with a Nyström approximation Williams and Seeger (2001) trained through stochastic gradient descent (SGD) Robbins and Monro (1951). This has been chosen, as it was identified as the best performing regressor among four other classical methods Rolff et al. (2022). The models were trained as specified in Rolff et al. (2022). One notable exception while training is the used sampling strategy. Here, we made sure that samples leading to the same gaze event are placed inside the same train, test, or validation dataset. To train the models, we utilize the DGaze Hu et al. (2020), FixationNet Hu et al. (2021) and EGTEA Gaze+ Li et al. (2018) datasets. In addition, we use some artificial prediction strategies to evaluate the proposed metrics on synthetically generated predictions. For those, we employ: mean time-to-saccade (mean), zero prediction (zero), maximum time-to-saccade (max), and random time-to-saccade prediction (rand). A more extensive evaluation of those can be found in the appendix.

Table 1 shows the measured results of the predictions on the DGaze Hu et al. (2020), FixationNet Hu et al. (2021) and EGTEA Gaze+ Li et al. (2018) datasets. While close to previous literature Rolff et al. (2022), the results are still slightly different due to the different sampling method. However, it is also evident, that this results in a higher overestimation rate, as the predictor can not estimate the correct time-to-saccade for most data samples. Moreover, the consistency of the SGD predictor is not as optimal as the average prediction. This is expected, as the average predictor reports very consistent results by predicting the

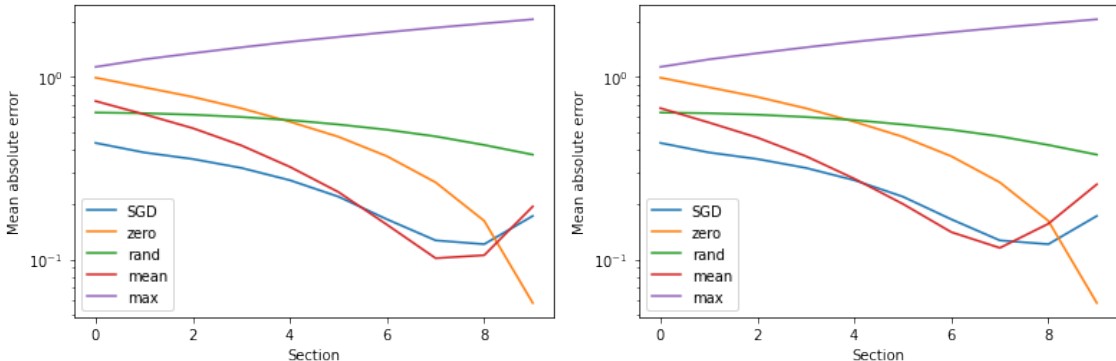

Figure 2: Error of different predictors on different sections as explained in Sec. 3. We divide all time-to-saccade sequences into 10 different sections of equal length to estimate the mean square error on the DGaze Hu et al. (2020) (left) and FixationNet Hu et al. (2021) (right) datasets.

mean value for every sample. It can also be seen that the underestimation error reports much lower results for the SGD predictor when compared to the average time-to-event. This is consistent with the underestimation rate, as the predictor underestimates less than the average predictor, making it more useful for real-world applications. Here we assume underestimations to be more of a problem than overestimations due to their ability to trigger downstream methods with the user being aware of them. In contrast, an overestimation can be mitigated through the utilization of data samples from the eye-tracker. Fig. 2 shows the evaluation of the 5 different baseline predictor models along with the SGD predictor over multiple sections of all sequences. Here, it can be seen that the SGD predictor outperforms all baselines most of the time, except for a brief range 20-30% of the length before the actual event, where it is outperformed by the mean absolute error. This indicates that the SGD tends to do better than the other predictor, but eventually fails shortly before the actual event. We also performed additional evaluations, which can be found in the appendix due to space restrictions.

## 5. Conclusion

In this paper, we proposed a new sampling strategy that lets us take the sequential information of gaze data for time-to-saccade prediction into account. This enabled us to define multiple new metrics capturing the consistency and duration of time-to-saccade predictors, as well as capturing the overall behavior of them over different parts of time-to-saccade sequences. To evaluate these, we use the state-of-the-art time-to-saccade predictor and compared it against a simple average baseline. However, we also expect future work on this topic, especially overestimation and underestimation evaluation, as they currently just evaluate the average over- and underestimation over the whole sequence but do not take the prediction strategy of a proposed model into account.

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

## Appendix A. Extended evaluation

To provide a more detailed evaluation on the proposed metrics of our paper, we also evaluate against five different baseline predictors, namely:

- Zero predictor (zero): Predicts an event for every step by reporting a time-to-event of zero. This should result in high underestimation and low overestimation rates. Furthermore, it should have a good but not zero consistency.

- Maximum predictor (max): This predicts the maximum time-to-event. This should result in high overestimation and low underestimation rates. Similar to the zero predictor, it should also have a good consistency. However, we also expect this to perform the worst out of all predictors on the mean square and mean absolute error.

- Random predictor (rand): Predicts a uniformly distributed random event length at the start of a sequence and consistently reduces the time-to-saccade by the update rate of the eye-tracker. Once, we predict a time-to-event below zero, we just report that the event is going to happen every step. We expect this predictor to have a mixed overestimation and underestimation rate dependent on the distribution of the time-to-event and an excellent consistency, due to its definition.

Using those predictors, we measure how the proposed metrics behave on the DGaze Hu et al. (2020), FixationNet Hu et al. (2021) and EGTEA Gaze+ Li et al. (2018) datasets.

First, Tab. 2 shows the evaluation of the zero predictor, which will report a time-to-event of zero. As expected, it is evident that the predictor underestimates every prediction, which is also shown through the underestimation rate. This also results in a high underestimation error, as the full sequence underestimates the actual target. Second, Tab. 3 shows the evaluation of the random predictor. As expected, this predictor has a much lower underestimation rate and very high consistency. It does not reach a 0.5 overestimation and underestimation rate, which is due to the uniform sampling not reflecting the general distribution of the data. At last, Tab. 4 shows the maximum predictor. Here, it is shown that the predictor does not produce any underestimations and thus has an excellent underestimation error. However, this also results in the highest average time-to-saccade errors, meaning that it does not well in its overall prediction. Moreover, Fig. 3 and Fig. 4 show the overestimation and underestimation rate using 10 sections to visualize the behavior of the overestimation and underestimation over time. As expected, the zero and maximum predictors have the highest over- and underestimation rate across the sequence lengths. Whereas, the random predictor is consistently at a 0.6 overestimation and 0.4 underestimation rate. It can also be inferred that the mean and SGD predictors tend to overestimate as the sequence reaches the event.

Table 2: Results of the zero predictor using the metrics described in Sec. 3 of the main paper and the mean square error (mse) and mean absolute error (mae). A lower error is preferred in all cases.

| Metric / Dataset | DGaze | FixationNet | EGTEA |
|---|---|---|---|
| mse↓ | $0.5168 \text{ s}^2$ | $0.6695 \text{ s}^2$ | $0.5168 \text{ s}^2$ |
| mae↓ | $0.5434 \text{ s}$ | $0.6408 \text{ s}$ | $0.5434 \text{ s}$ |
| avg. time-to-saccade mse↓ | $0.2089 \text{ s}^2$ | $0.3088 \text{ s}^2$ | $0.2000 \text{ s}^2$ |
| avg. time-to-saccade mae↓ | $0.4066 \text{ s}$ | $0.3831 \text{ s}$ | $0.3733 \text{ s}$ |
| underestimation mse↓ | $0.2089 \text{ s}^2$ | $0.3088 \text{ s}^2$ | $0.2000 \text{ s}^2$ |
| underestimation mae↓ | $0.4066 \text{ s}$ | $0.3831 \text{ s}$ | $0.3733 \text{ s}$ |
| overestimation rate↓ | 0.0 | 0.0 | 0.0 |
| underestimation rate↓ | 1.0 | 1.0 | 1.0 |
| consistency↓ | 1.0 | 1.0 | 1.0 |

Table 3: Results of the random predictor using the metrics described in Sec. 3 of the main paper and the mean square error (mse) and mean absolute error (mae). A lower error is preferred in all cases.

| Metric / Dataset | DGaze | FixationNet | EGTEA |
|---|---|---|---|
| mse↓ | $0.4585 \text{ s}^2$ | $0.6805 \text{ s}^2$ | $0.7973 \text{ s}^2$ |
| mae↓ | $0.5384 \text{ s}$ | $0.6526 \text{ s}$ | $0.7066 \text{ s}$ |
| avg. time-to-saccade mse↓ | $0.5097 \text{ s}^2$ | $0.7877 \text{ s}^2$ | $0.9966 \text{ s}^2$ |
| avg. time-to-saccade mae↓ | $0.5742 \text{ s}$ | $0.1017 \text{ s}$ | $0.8003 \text{ s}$ |
| underestimation mse↓ | $0.0653 \text{ s}^2$ | $0.1017 \text{ s}^2$ | $0.0575 \text{ s}^2$ |
| underestimation mae↓ | $0.1306 \text{ s}$ | $0.1620 \text{ s}$ | $0.1006 \text{ s}$ |
| overestimation rate↓ | 0.62 | 0.62 | 0.72 |
| underestimation rate↓ | 0.38 | 0.38 | 0.28 |
| consistency↓ | 0.24 | 0.25 | 0.20 |

Table 4: Results of the maximum predictor using the metrics described in Sec. 3 of the main paper and the mean square error (mse) and mean absolute error (mae). A lower error is preferred in all cases.

| Metric / Dataset | DGaze | FixationNet | EGTEA |
|---|---|---|---|
| mse↓ | 2.7385 $s^2$ | 3.9040 $s^2$ | 4.1814 $s^2$ |
| mae↓ | 1.6050 s | 1.9092 s | 1.9899 s |
| avg. time-to-saccade mse↓ | 2.9792 $s^2$ | 4.3473 $s^2$ | 4.7266 $s^2$ |
| avg. time-to-saccade mae↓ | 1.7134 s | 2.0669 s | 2.1601 s |
| underestimation mse↓ | 0.0000 $s^2$ | 0.0000 $s^2$ | 0.0000 $s^2$ |
| underestimation mae↓ | 0.0000 s | 0.0000 s | 0.0000 s |
| overestimation rate↓ | 1.0 | 1.0 | 1.0 |
| underestimation rate↓ | 0.0 | 0.0 | 0.0 |
| consistency↓ | 1.0 | 1.0 | 1.0 |

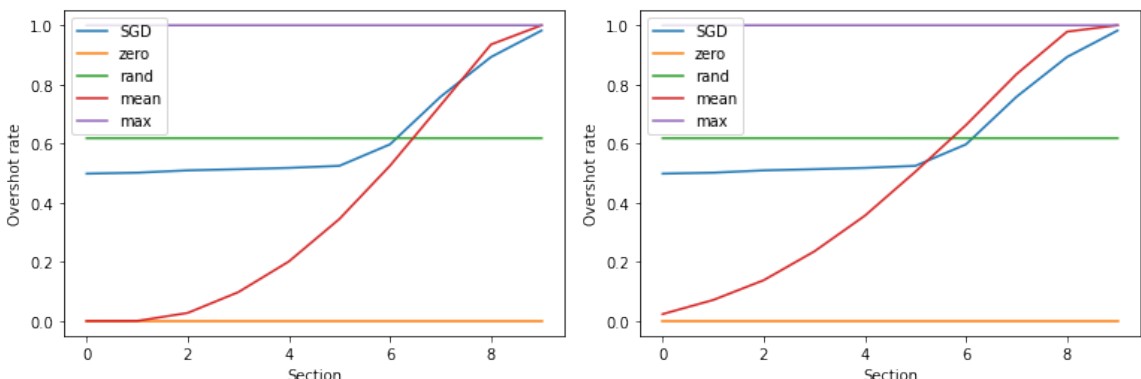

Figure 3: Overestimation rate calculated over 10 sections on the DGaze Hu et al. (2020) and FixationNet Hu et al. (2021) datasets.

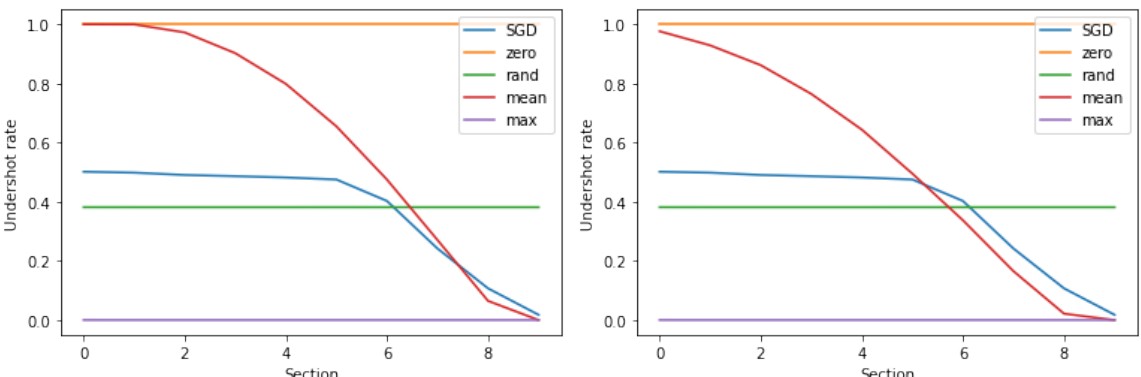

Figure 4: Underestimation rate calculated over 10 sections on the DGaze Hu et al. (2020) and FixationNet Hu et al. (2021) datasets.

