# OpenReview forum: "Metrics for Time-to-Event Prediction of Gaze Events"
_NeurIPS.cc/2022/Workshop/GMML — Gaze Meets ML 2022 Poster_

### Official Review · Reviewer_X8UR · 2022-10-09
**Useful contribution**

**Rating:** 7
**Confidence:** 3

**Review:**

**General considerations**

Authors propose a novel sampling strategy that takes into account the sequential information of the time-to-saccade in gaze; as a derivation, they can provide novel error metrics that allow for better understanding the performance of a predictor.

A single predictor is compared with a simple baseline. The work could benefit from presenting additional predictors. Also, a large number of metrics is given, hence it could be beneficial to provide recommendations on how to select the "best model" for a specific application in case results might be mixed.

Overall the contribution is rather limited but might be beneficial for the specific or related fields of application.

**Clarity**

The paper is very well written. Related work is discussed properly. Contributions and claims are clearly stated.

The reference to "real world evaluation" in the paper title is somewhat misleading and the authors could propose an alternative which better describes the main paper contribution.

**Originality and significance**

The work seems novel, since sequential properties of the phenomenon were never exploited previously, which is relevant in the specific application. However, I am not fully aware of the literature in the specific field.

The proposed metrics enrich the possibilities for predictor performance interpretation.

---

### Official Review · Reviewer_s29w · 2022-10-15
**New metrics for time-to-saccade prediction**

**Rating:** 1
**Confidence:** 5

**Review:**

The author proposed to using updated sampling to evaluate the time-to-saccade problems. Instead of random sampling as proposed in existing studies, the author proposed to correct the sampling by making sure the same events occur in training/testing/validation datasets, so as to get a more consistent evaluation. The work seems to be fairly trivial and the contribution is not clear.

---

### Official Review · Reviewer_Lt3V · 2022-10-17
**This work introduces new metrics and sampling strategies to evaluate the time-to-saccade prediction. While important in understanding the drawbacks of state-of-the-art approaches, the reasoning behind using each metric is insufficient and lacks mathematical representation, making it hard to follow.**

**Rating:** 5
**Confidence:** 2

**Review:**

**Strengths:**

* The authors provided sufficient motivations for the need for new metrics to evaluate time-to-saccade predictions.

* This work explores different metrics to evaluate the time-to-saccade prediction problem while considering the temporal characteristics of gaze data.

*  The introduced metrics provide more intuition and explanation for the reported results of an approach. In an evaluation study, the authors compared multiple methods using their metrics, providing crucial information about the performance across the temporal domain.

**Weaknesses:**

The metrics lack mathematic representations, which make them hard to follow. Without the strict mathematical expression, the reader can get confused and could end up evaluating a completely different property.

*  The authors have introduced a new sampling strategy that breaks the gaze time sequence at the time of events and derives specific properties. In the paper (77-80), they claimed "... that the first time-to-event value in the sequence is always equal to the length of the whole sequence and the last time-to-event value is always one step before the desired event. Another advantage is that the time-to-event of such a sequence is always strictly monotonically decreasing, with the rate being depended on the update rate of the eye-tracker." I find this reasoning very hard to follow and would recommend the authors provide a diagram explaining their sampling strategy, similar to Figure 1 of Rolff et al.

* I find the mathematical expression of consistency a bit confusing. If $\{ p_i, p_{i+1}\}$ are the predicted times to saccade and $f_{i+1}$ is the update rate then shouldn't consistency be $c_j \propto \sum | |p_{i+1}-p_i| - f_{i+1}^{-1} |$. In other words, in $p$ represents time and $f$ represents frequency, then $p$ should be compared with $f^{-1}$ not $f$.

*  The metrics defined for undershooting error and sectioning are hard to follow. The authors can significantly improve the work if they provide the mathematical expression for each of the metrics.

---

### Meta-Review · Area_Chair_UfiT · 2022-10-20

**Recommendation:** Accept (Poster)
**Confidence:** 4

**Metareview:**

Authors propose a new sampling strategy for evaluating time-to-saccade events in gaze. The paper introduces different metrics with temporal properties. Evaluation of the proposed methods and comparisons shows the performances of the proposed sampling strategy.

The reviewers remarked that there were certain aspects of the paper that needed more clarity and that some of the work, while important, was preliminary.

This is a good paper for the workshop, and I recommend an acceptance.

---

### Decision · Program_Chairs · 2022-10-20

Accept (Poster)